# Phytochemical Investigation of Three *Cystoseira* Species and Their Larvicidal Activity Supported with In Silico Studies

**DOI:** 10.3390/md21020117

**Published:** 2023-02-10

**Authors:** Shaza H. Aly, Ahmed M. Elissawy, Dina Salah, Nawal Abdulaziz Alfuhaid, Ola H. Zyaan, Hany I. Mohamed, Abdel Nasser B. Singab, Shaimaa M. Farag

**Affiliations:** 1Department of Pharmacognosy, Faculty of Pharmacy, Badr University in Cairo, Cairo 11829, Egypt; 2Department of Pharmacognosy, Faculty of Pharmacy, Ain Shams University, Cairo 11566, Egypt; 3Centre of Drug Discovery Research and Development, Ain Shams University, Cairo 11566, Egypt; 4Department of Physics, Faculty of Science, Ain Shams University, Cairo 11566, Egypt; 5Department of Biology, College of Science and Humanities in Al-kharj, Prince Sattam Bin Abdulaziz University, Al-kharj 11942, Saudi Arabia; 6Department of Entomology, Faculty of Science, Ain Shams University, Cairo 11566, Egypt; 7Chemistry Department, Faculty of Science, Benha University, Benha 13518, Egypt

**Keywords:** acetylcholinesterase, brown algae, biological activity, *Culex pipiens*, *Cystoseira*, larvicidal activity, molecular docking

## Abstract

*Culex pipiens* mosquitoes are transmitters of many viruses and are associated with the transmission of many diseases, such as filariasis and avian malaria, that have a high rate of mortality. The current study draws attention to the larvicidal efficacy of three methanolic algal extracts, *Cystoseira myrica*, *C. trinodis*, and *C. tamariscifolia*, against the third larval instar of *Cx. pipiens*. The UPLC-ESI-MS analysis of three methanol fractions of algal samples led to the tentative characterization of twelve compounds with different percentages among the three samples belonging to phenolics and terpenoids. Probit analysis was used to calculate the lethal concentrations (LC_50_ and LC_90_). The highest level of toxicity was attained after treatment with *C. myrica* extract using a lethal concentration 50 (LC_50_) of 105.06 ppm, followed by *C. trinodis* (135.08 ppm), and the lowest level of toxicity was achieved by *C. tamariscifolia* (138.71 ppm) after 24 h. The elevation of glutathione-S-transferase (GST) and reduction of acetylcholine esterase (AChE) enzymes confirm the larvicidal activity of the three algal extracts. When compared to untreated larvae, all evaluated extracts revealed a significant reduction in protein, lipid, and carbohydrate contents, verifying their larvicidal effectiveness. To further support the observed activity, an in silico study for the identified compounds was carried out on the two tested enzymes. Results showed that the identified compounds and the tested enzymes had excellent binding affinities for each other. Overall, the current work suggests that the three algal extractions are a prospective source for the development of innovative, environmentally friendly larvicides.

## 1. Introduction

Human health is one of the most alarming effects of global warming in a variety of intricate ways. The population is significantly impacted by the changed spatial distribution of various infectious disease vectors, such as mosquitoes [1]. Changes in temperature may increase the possibility of the transmission of numerous mosquito-borne diseases, including the West Nile virus, dengue fever, and avian malaria. This is due to the fact that temperature affects both the pathogenic organisms’ and vector species’ life-cycle dynamics. Warmer temperatures increase mosquito reproduction and accelerate the development of the microorganisms they distribute [2]. In urban areas, female *Culex pipiens* feed on a range of vertebrate hosts, which may help to promote the spread of West Nile virus among birds and occasionally to populations of humans and other mammals [3]. Despite being extensively dispersed now, *Cx. pipiens* are native to Africa, Asia, the Middle East, and Europe [4]. Frequent vector management is the primary strategy for preventing and controlling the spread of mosquito-borne diseases [5]. In the past decades, conventional chemical insecticides have been created to combat diseases spread by mosquitoes [6]. Chemical pesticides are hazardous to humans and have a harmful impact on the environment and other living things; moreover, the unwise use of synthetic insecticides leads to resistance in mosquito species [7,8]. Researchers are searching for pest control alternatives that have fewer negative side effects. As a result, attention has been drawn to research on natural products that tries to identify the active components of natural insecticide agents that are secure and target-specific. Also, developing larvicides from marine sources might be another technique of mosquito control. Success stories in the search for pesticides with a natural origin have been documented [9,10,11,12].

Algae are organisms that resemble plants and have photosynthetic pigments in their cells. They are most commonly found in freshwater, marine, and wastewater ecosystems and range in size from microalgae to macroalgae. Algae are regarded as a rich source of a variety of biologically active molecules [13]. They also create a number of secondary metabolites with a wide range of chemical compositions, some of which are known insecticides. Secondary metabolites extracted from algae have been demonstrated to have a substantial effect on mosquito larvae [14].

Marine macroalgae, generally known as seaweed, presented a new tactic in the field of pest control [15]. Seaweeds are distributed in shallow coastal waters, estuaries, and intertidal and deep-sea regions [16]. They are typically categorized into three large pigmentation-based groups: Chlorophyta (green algae), Phaeophyta (brown algae), and Rhodophyta (red algae) [17]. Seaweeds, with their great chemical diversity, are utilized as a foodstuff for human and animal food, fertilizers, cosmetics, and medicinal products [18,19]. Seaweed extracts have proven competence for controlling and repelling various insect pests such as cereal aphids, *Schizaphis graminum* [20], the cotton stainer bug, *Dysdercus cingulatus* [21], mosquitoes [22,23], the tomato moth, *Tuta absoluta* [24], the termite, *Microtermes obesi* [25], maize weevil, *Sitophilus zeamais* [26] and the Asian citrus psyllid, *Diaphorina citri* [27]. Some studies documented the insecticidal effects of seaweeds of Egyptian origin as *Caulerpa prolifera*, *Caulerpa serrulata*, *Jania rubens*, *Nitophyllum punctatum Padina pavonica*, Chara vulgaris, *Parachlorella kessleri*, *Ulva intestinalis* and *Cladophora glomerata* [28,29].

*Cystoseira* is a widespread brown algae genus found throughout the Mediterranean region. Numerous substances have been identified from various species of the Mediterranean brown algae of the genus *Cystoseira*, including terpenoids, alkaloids, polysaccharides, and steroids; however, very few investigations on the pharmacological characteristics of these substances and different *Cystoseira* species have been published [30,31,32,33,34]. It is crucial to research the bioactive ingredients of seaweeds and assess their efficacy for mosquito control. Information on the structure-activity relationships of active chemicals that are responsible for the killing action can be obtained by understanding the chemical components of seaweed [14]. Recently, the dependence on secondary metabolites from natural sources has been in great demand to control various ailments and defense against several conditions [35,36,37,38,39,40].

Based on the prospective findings, the objective of this study is to investigate the chemical composition of the methanol fraction of three *Cystoseira* species, *C. myrica*, *C. trinodis*, and *C. tamariscifolia*, using UPLC/MS analysis along with a screening of the three species to generate eco-friendly larvicide algal extracts and assess their bioactivity against *Cx*. *pipiens* larvae. This is besides the correlation of their activity through molecular docking of the major constituents.

## 2. Results and Discussion

### 2.1. UPLC-ESI-MS Analysis for Characterization of Cystoseira myrica, C. trinodis, and C. tamariscifolia Methanol Fraction

Metabolic profiling of three algal species, *C. myrica*, *C. trinodis*, and *C. tamariscifolia* methanol fraction, was achieved using UPLC/ESI/MS analysis (Appendix A). Twelve compounds with different concentrations were annotated in the methanol fraction of the three *Cystoseira* species. Most of them belonged to the class of terpenoids and flavonoids (Table 1). Compounds were tentatively identified based on the mass of the molecular ion peaks and comparison with bibliographic references, as shown in (Table 1). On the one hand, a major meroditerpenoid with molecular formula C_18_H_26_O_2_ (**1**) and with a mass ion peak at *m*/*z* 273 was identified in *C. myrica* and *C. tamariscifolia*, and it was previously reported from *C. baccata* [41]. Acyclic diterpene (2*E*, l0*E*)-1-hydroxy-6,13-diketo-7- methylene-3,11,15-trimethylhexadeca-2,l0,l4-triene (**3**) with mass ion peak at *m*/*z* 317 identified in *C. tamariscifolia* and previously reported from *C. crinite* [42]. A phlorotannin with a mass ion peak at *m*/*z* 517 and molecular formula C_24_H_22_O_13_ was identified as bifuhalol hexacetate (**5**) in *C. myrica*. Oxocrinol (**9**) was identified as a linear terpenoid with a mass ion peak at *m*/*z* 223 and molecular formula C_14_H_24_O_2_ in *C. tamariscifolia*, and it was previously reported from *C. crinite* [43]. Hydroazulene diterpene cystoseirol monoacetate (**10**) with mass ion peak at *m*/*z* 397 and molecular formula C_22_H_36_O_5_ identified in the three algal extracts, and it was previously isolated from *C. myrica* [44]. Moreover, α-linolenic acid (**12**) with a mass ion peak at *m*/*z* 277 was identified in *C. tamariscifolia*. Linolenic acid was previously identified in *C. compressa* and *C. indica* extracts [31,45].

It is worth noting that Phenolic compounds are also identified in the tested methanol fractions as compound (**2**) myricetin and compound (**6**) quercetin, which had been identified previously using the HPLC technique from family Sargassaceae in *Sargassum latifolium* and *S. wightii* [46,47]. On the other hand, some tentatively identified constituents, including luteolin (**4**), cyanidin-3-*O*-glucoside (**7**), along with 7,8-methylenedioxycoumarin (**8**), and apigenin (**11**).

Our results revealed the presence of variable secondary metabolites with different percentages of the methanol fraction of the three species under investigation. The chemical structures of the major tentatively-identified compounds are presented in (Figure 1).
marinedrugs-21-00117-t001_Table 1Table 1Metabolite profiling of *Cystoseira myrica*, *C. trinodis*, and *C. tamariscifolia* Methanol fraction via UPLC-ESI-MS in the negative ion mode.No.R_t_ (min)Compound Name[M − H]^−^(*m*/*z*)MolecularFormulaRelative Amount (%)Ref.*C. myrica**C. trinodis**C. tamariscifolia*11.29Compound 7273.00C_18_H_26_O_2_8.10-7.65[41]21.56Myricetin317.05C_15_H_10_O_8_
--9.99[48]31.75(2*E*, l0*E*)-1-hydroxy-6,13-diketo-7-methylene-3,11,15-trimethylhexadeca-2, l0, l4-triene317.00C_20_H_30_O_3_
--3.45[42]42.04Luteolin-7- glucoside447.05C_21_H_20_O_11_--2.36[49]53.67Bifuhalol hexacetate517.00C_24_H_22_O_13_
5.83--[50]64.60Quercetin301.00C_15_H_10_O_7_0.360.18-[51]75.05Cyanidin-3-O- glucoside485.05C_21_H_21_O_11_^+^48.2710.06-[52]85.587,8-Methylenedioxycoumarin 189.10C_10_H_6_O_4_
--13.87[53]97.32Oxocrinol223.10C_14_H_24_O_2_--6.50[43]107.86Cystoseirol monoacetate397.05C_22_H_36_O_5_
1.290.491.21[44]118.73Apigenin269.20C_15_H_10_O_5_
--1.40[54]1223.68α-Linolenic acid 277.00C_18_H_30_O_2_
--4.79[55]R_t_: Retention time recorded for each compound.

### 2.2. Larvicidal Bioassay

The bioassay test of the methanol fraction of *C. myrica*, *C. trinodis*, and *C. tamariscifolia* was conducted on the third instar larvae of *Cx. pipiens*. The mortality of *Cx. pipiens* larvae was calculated at 24 h, 48 h, and 72 h post-treatment (Table 2). The results demonstrated that the toxicity of the tested extracts significantly increased with time, extract concentration, and length of exposure time. With higher concentrations and longer exposure times, there was a higher percentage of larval mortality. The mortality of *Cx. pipiens* larvae started on the first day of exposure and continued until the third day. The highest level of toxicity was attained by *C. myrica*, followed by *C. trinodis* at LC_50_ 105.06 and 135.08 ppm, respectively. The lowest level of toxicity was achieved by *C. tamariscifolia* at LC_50_ (138.71 ppm) at 24 h post-treatment (Table 2).

The activity of the tested extracts was arranged as follows: *C. myrica* > *C. trinodis* > *C. tamariscifolia* (Table 2). In general, *C. tamariscifolia* showed low convergent toxicity, while *C. myrica* and *C. trinodis* displayed adequate action against *Cx. pipiens* larvae. The low slope values suggested that the tested population of *Cx. pipiens* larvae are uniform.

Similar larvicidal activities were reported from different brown algae extracts against *Cx. pipiens* larvae as methanol extract of *Sargassum dentifolium*, *Dictyota dichotoma*, and *Padina boryana* exhibited LC_50_ values of 306.86, 266.85, and 295.52 ppm, respectively, after 24 h post-exposure [9]. Another study reported the larvicidal activity of the ethanolic extract of *Cystoseira barbata* against larvae of *Aedes albopictus* [56]. The MgO-NPs of *Cystoseira crinita* extract were tested against the house fly *Musca domestica* at different concentrations. It showed the highest mortality percentages, 99.0%, 95.0%, 92.2%, and 81.0% for 1st, 2nd, and 3rd instars’ larvae and pupa of *M. domestica*, respectively, at 10 μg/mL MgO-NPs [57]. The three tested extracts induced abnormalities in the treated larvae and also in the pupae that resulted from treated larvae (Appendix A).

### 2.3. Biochemical Analysis

Insects synthesize many detoxifying enzymes, including esterases, oxidases, and reductases, to react with and detoxify a variety of invasive pesticides [58]. We investigated the activity of two different enzymes, glutathione-S-transferase (GST) and acetylcholinesterase (AChE), to gain an insight into the mechanisms involved in the 3rd larval instar of *Culex pipiens*. Through inhibition of AChE, the essential enzyme in regulating the level of acetylcholine in cholinergic synapses of insects will result in permanent neural excitation/stimulation, paralysis, ataxia, and eventually death [59,60]. The obtained results showed the activity of the GST enzyme significantly increased due to treatment with *C. myrica*, *C. trinodis*, and *C. tamariscifolia*. While the activity of AChE was decreased 24 h and 48 h post-treatment with the tested extracts as compared to the untreated group, as shown in (Table 3). Our results were in accordance with the previous reports that confirmed the role of GST and AChE in the detoxification process of the extract. Abdel Haleem et al. (2022) reported an increase in the intracellular glutathione content when the larvae were treated with different algal extracts [9]. Also, our findings are consistent with those of Huang et al. (2013), which discovered an increase in intracellular glutathione concentration after treating larvae with a polyphenolic-rich extract [54,61].

### 2.4. Determination of Total Proteins, Total Lipids, and Total Carbohydrates

As proteins, carbohydrates, and lipids are the three most important components required for larval growth and development, the decline in these three components led to impaired larval development, which eventually led to larval death.

The biochemical changes in the whole-body tissue of *Cx. pipiens* larvae (carbohydrate, protein, and lipid) are shown in (Figure 2). All tested extracts showed a significant reduction in protein, lipids, and carbohydrate contents of treated larvae compared to untreated larvae, which may impair the survival and development of the larvae. Our results were in accordance with previous reports that reported that larvicides have a deleterious impact on larval growth and development, causing alterations in their metabolic and biochemical processes [62].

### 2.5. In Silico Studies

#### 2.5.1. Binding Mode with AChE Enzyme

The main compounds identified in the methanol fraction of *C. myrica*, *C. trinodis* and *C. tamariscifolia*, and λ-cyhalothrin (control) were docked separately into the active pocket of AChE (PDB ID: 6XYS), to investigate the nature of binding with the crucial enzymes and discover some insights about the mode of action.

The meroditerpenoid with molecular formula C_18_H_26_O_2_ (**1**) and compound (**2**) myricetin showed good binding affinities to AChE without forming any hydrogen bonds (Appendix A); only exposure to the receptor controls their binding effectiveness. Moreover, compound (**3**), with its long carbon chain, is bound strongly to Trp321 via H-arene interactions (Figure 3). While compounds (**5**) and (**10**) showed a high binding affinity with a binding energy of −7.26 and 7.28 Kcal/mol, respectively, but without forming any hydrogen bonds. Also, compound (**9**) showed high binding energy equal to −7.28 Kcal/mol (Table 4). In addition, α-linolenic acid (**12**) (BE = −7.97 Kcal/mol) adapted to the pocket through the formation of ion-H (Figure 4B) and H-arene (Figure 4E) interactions with Trp83 and Trp321 residues, respectively. The reference insecticide, λ-cyhalothrin, exhibited a very close binding behavior to (**2**) and (**10**) (Figure 3 and Figure 4).

The 2D-binding interaction profile for compounds **4**, **7, 8,** and **11** with the active pocket of the AChE enzyme are represented in (Appendix A).

Collectively, the activity of *C. myrica* fraction toward *Cx. pipiens* could be mainly attributed to the binding of compounds (**1** and **5**) with the AChE enzyme. For the *C. trinodis* fraction, compound (**10**), although its small concentration, shows high binding affinity. Regarding the *C. tamariscifolia* fraction, compounds (**3**, **9**, and **12**) furnish the activity, and maybe in all extracts, each compound possesses a partial effect.

#### 2.5.2. Binding Mode with GST Enzyme

The compounds identified in the methanol fraction of *C. myrica*, *C. trinodis*, *C. tamariscifolia*, and λ-cyhalothrin were subjected to docking into the pocket of GST protein (PDB ID: 1M0U). This study may help in predicting the mode of action of these derivatives as insecticides and discovering their abilities to bind with crucial enzymes.

The results for docking and the GST-binding parameters are depicted in (Figure 5 and Figure 6) and (Table 4). Compound (**3**) showed a comparable binding effect with the reference (BE = −5.75 Kcal/mol), although it could not form any type of interaction with the residues of the GST pocket (Figure 5A,D). This may be due to its long carbon chain that accurately fits the size of the GST pocket. Also, compound (**9**) has a similar behavior to compound **(3)** because of structural similarity; it could form one weak hydrogen bond with Ser110 (Figure 5F). Compound (**5**) binds with the GST pocket by forming one hydrogen bond (Figure 5E). Besides, compound (**10**) possesses two hydrogen bonds with two crucial amino acid residues and a value of binding energy (−6.06 Kcal/mol). Three hydrogen bonds control the binding of compound (**20**) with the pocket of GST (Figure 6B,E). It shares the formation of H-bonds with two crucial residues, Arg145 and Tyr208. The reference insecticide, λ-cyhalothrin exhibited only pi-pi interaction with Phe55 (Figure 6F). It can be inferred from the docking data that Phe55, Gln96, Ser110, Arg145, and Tyr208 represent the crucial amino acid residues in the active pocket of GST.

The 2D-binding interaction profile for compounds **1**, **2**, **4**, **7**, **8**, and **11** with the active pocket of GST enzyme are represented in (Appendix A).

## 3. Materials and Methods

### 3.1. Plant Material

The three algae samples, *Cystoseira myrica*, *C. trinodis*, and *C. tamariscifolia*, were collected in October 2020 from the Gulf of Suez shores, Ras Sedr city, Egypt 29°50′19.7″ N and 32°37′36.8″ E. A voucher sample was identified according to Ibraheem et al. 2014 [63] and was deposited at the Pharmacognosy Department, Faculty of Pharmacy, Badr University in Cairo under codes BUC-PHG-CM-3, BUC-PHG-CT-4 and BUC-PHG-CT-5, respectively. The collected algal samples were brought to the laboratory in sterilized plastic bags containing seawater to prevent evaporation. They were also cleaned of epiphytes and rock and sand debris and then given a quick freshwater rinse to remove any adhered surface salts or residues. Photos of the three samples are included in (Appendix A).

### 3.2. Preparation of the Plant Extract

The grounded air-dried powder of each sample (100 g), *Cystoseira myrica*, *C. trinodis*, and *C. tamariscifolia* were extracted with pure methanol by maceration method three times at room temperature (3 × 500 mL) followed by concentration by evaporation in vacuo at low temperature (45 °C) to yield a dark brown residue 18.2, 26, and 30 g, respectively [27].

### 3.3. UPLC-ESI-MS Analysis

UPLC/MS analysis was performed at the Centre of Drug Discovery Research and Development, Department of Pharmacognosy, Faculty of Pharmacy, Ain Shams University, Egypt, using Waters^®^TQD UPLC-MS with an ESI source using waters^®^ acquity UPLC RP-C18 column, (100 × 2 mm, ID), and particle size of 1.7 µm, with an integrated pre-column. A gradient of water and acetonitrile (with 0.1% formic acid) was applied from 2% to 100% acetonitrile. The flow rate was 1 or 0.5 mL/min, and one run took 35 min. The MS was operated at −10 V for ESI-, 240 °C source temperature, and high purity nitrogen as a sheath and auxiliary gas at flow rates of 80 and 40 (arbitrary units), respectively. The injection volume was 5 μL. The spray voltage was 4.48 kV, the tube lens voltage was 10.00 V, and the capillary voltage was 39.6 V. A full scan mode was adjusted in the mass range of 100–2000 *m*/*z* [64].

Tentative metabolite assignments were done based on the MS data (in the negative ionization mode) by comparison with the previously reported compounds from the genus and the family [41,42,43,44,50,65,66] alongside online public databases. Data acquisitions and analyses were executed by XcaliburTM 2.0.7 software (Thermo Scientific, Karlsruhe, Germany).

### 3.4. Insect Culture

Egg rafts of *Cx. pipiens* were obtained from colonies maintained at the Research and Training Center of Vector Disease, Ain Shams University, Cairo. The eggs were transferred to white metal enamel plates containing dechlorinated tap water and reared in the insectary for 3rd instar larvae collection for further bioassay experiments. Mosquitoes were maintained at 25 ± 2 °C and 75% RH under a 12:12 L/D photoperiod. Newly-hatched larvae were fed a diet of tetramine. Pupae were transferred to cups containing dechlorinated tap water and placed in rearing cages (40 × 30 × 25 cm), where adults emerged. Cotton pieces soaked in a 10% sucrose solution were used to feed adults. Females were provided with a pigeon placed on the mesh of the cage for blood feeding (Appendix A). This research paper was approved by the research ethics committee from the Faculty of Science, Ain Shams University (ASU-SCI/ENTO/2023/1/2).

### 3.5. Larvicidal Bioassay

An amount of 10 g of each crude extract of *Cystoseira myrica*, *C. trinodis*, and *C. tamariscifolia* was dissolved in 100 mL of ethanol as a 10 percent stock solution. Distilled water was used to prepare five different concentrations of each extract (50, 100, 150, 200, and 250 ppm). Twenty-five larvae of the third instar were placed in plastic cups containing five different concentrations of each extract, while in the control, ethanol was used as calculated from the highest concentration preparation [10,67]. The experiment was replicated three times. Mortality data were recorded after 24, 48, and 72 post-treatment. The mortality percentages were corrected using Abbott’s formula [68].

### 3.6. Biochemical Assay

Insects were smashed in a chilled glass Teflon tissue homogenizer. Homogenized samples were centrifuged at 8000 rpm for 15 min at 5 °C in a refrigerated centrifuge. Supernatants were kept in a deep freezer at −20 °C until use in biochemical assays.

#### 3.6.1. Estimation of Total Carbohydrates

By using the phenol sulfuric acid method, total carbohydrates were calculated, as stated in [69]. A 100 µL sample was mixed with 200 µL of sulfuric acid and 100 µL of phenol (5 g/100 mL). After 30 min of incubation, the resultant absorbance at 490 nm was measured.

#### 3.6.2. Estimation of Total Lipids

Total lipids were estimated by the method of Knight et al. (1972) using the phospho-vanillin reagent (20%) [70]. A 250 μL sample was mixed with 5 mL of sulfuric acid in a test tube and heated for 10 min in a boiling water bath. After cooling to room temperature, the sample was added to the phospho-vanillin reagent (6 mL). After 45 min, the absorbance of the obtained color was measured at 525 nm [71].

#### 3.6.3. Estimation of Total Proteins

Total protein was determined according to Bradford (1976) using a solution of Coomassie Brilliant Blue dissolved in 95% ethanol. 100 mL of the sample was mixed with 5 mL of Bradford reagent. The absorbance was measured at 595 nm [72].

### 3.7. Enzyme Activities

#### 3.7.1. Glutathione S-transferase (GST)

The activity of glutathione S-transferase (GST) was evaluated depending on Kao et al. (2016) method [73] CDNB (1- chloro-2,4-dinitrobenzene) was used as a substrate.

#### 3.7.2. Acetylcholine Esterase (AChE)

The activity of acetylcholinesterase (AChE) was measured in untreated and treated larval samples previously treated with LC_50_ of tested compounds using the substrate acetylcholine bromide (AChBr) [74].

### 3.8. Statistical Analysis

The results obtained were statistically calculated by using probit analysis [75] with the statistics program (LDP-line). The biochemical results were analyzed by one-way analysis of variance (One-way ANOVA). The means were compared by Tukey’s multiple comparisons tests. The significance of the differences between the tested and control groups was determined for every tested extract. A *p*-value of ≤ 0.05 was considered statistically significant.

### 3.9. In Silico Studies

The 3D-crystal structures for the macromolecules AChE (PDB ID: 6XYS) and GST (PDB ID: 1M0U) from *Drosophila melanogaster* [76,77] were retrieved from the protein data bank website (www.rcsb.org accessed on 1 December 2022). The major compounds identified in *C. myrica*, *C. trinodis*, and *C. tamariscifolia* methanol fraction and λ-cyhalothrin (reference insecticide) were docked separately into the active pocket of the receptors, AChE, and GST. All docking calculations and results visualizations were performed using a molecular operating environment (MOE) 2015.10 software package [78].

The preparation module in the MOE’s receptors was used to fix the protein structure for any missing atoms or residues and make any necessary adjustments. This process comprised applying partial charges through the MMFF94 (modified) forcefield, 3D-protonation, and minimizing the protein structures to a chosen gradient using default parameters. The binding pockets of AChE and GST were defined using the dummy atoms that were generated at the binding location using MOE’s site finder option. The docking algorithm was configured to use the triangle matcher placement approach with rigid receptor refinement. The triangular match algorithm was programmed to create 50 poses, but the induced fit refinement approach produced five poses. Furthermore, the default GBVI/WSA dG technique was used as a docking function in MOE [79]. Docking conformations with the lowest binding energies were chosen. Validation of the docking protocol was done by redocking ligands into the active pocket of AChE and GST proteins. Only those docking poses were considered successful whose root-mean-square deviation (RMSD) values were less than 2.0 Å [80].

## 4. Conclusions

The present study showed the UPLC/ESI/MS analysis of the methanol fraction of three algae samples, *Cystoseira myrica*, *C. trinodis*, and *C. tamariscifolia*, and revealed the presence of flavonoids and terpenoids as main compounds. Also, they showed promising larvicidal activity against third-instar larvae of *Cx. pipiens.* Moreover, the three fractions showed a reduction in acetylcholinesterase and an elevation in glutathione-S-transferase enzymes that enhance the detoxification process of *Cx. pipiens* larvae. The promising larvicidal activity of the tested samples was confirmed by in silico studies of the identified compounds towards two tested enzymes. Compounds, (2*E*, l0*E*) 1-hydroxy-6,13-diketo-7-methylene-3,11,15-trimethyl-hexadeca-2,l0,l4-triene, bifuhalol hexacetate, oxocrinol, cystoseirol monoacetate, and α-linolenic acid showed good binding affinities to the AChE enzyme. Besides, bifuhalol hexacetate and cystoseirol monoacetate showed good binding affinities to the GST enzyme. Further investigations are required to investigate the chemistry of the methanol extract of three species in-depth. To sum up, the findings of this study can be used to introduce a biodegradable and novel organic larvicide to control mosquitoes.

## Figures and Tables

**Figure 1 marinedrugs-21-00117-f001:**
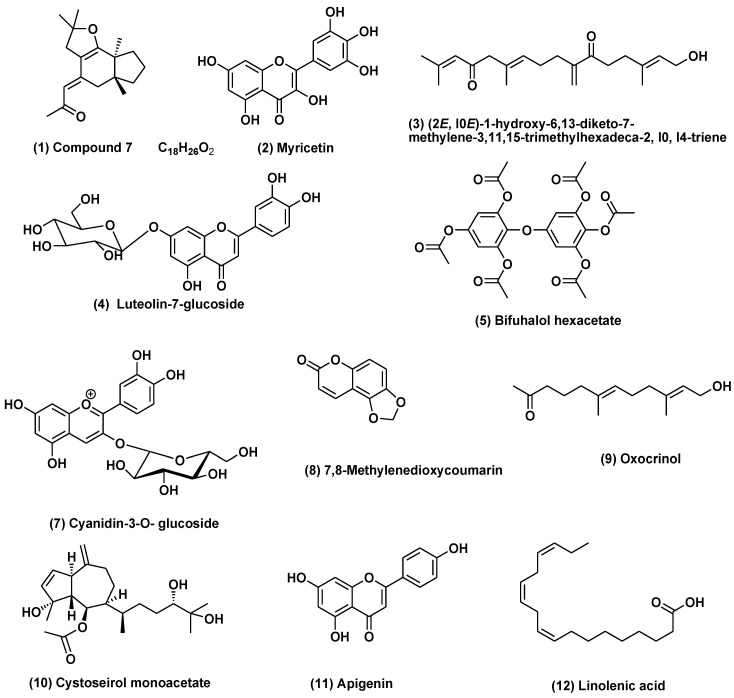
Structures of compounds identified in *Cystoseira myrica*, *C. trinodis*, and *C. tamariscifolia* methanol fraction using UPLC-ESI-MS in the negative ion mode. Compounds (**1**, **2**, **3**, **5**, **9**, **10**, and **12**) identified in different *Cystoseira* species and the family Sargassaceae.

**Figure 2 marinedrugs-21-00117-f002:**
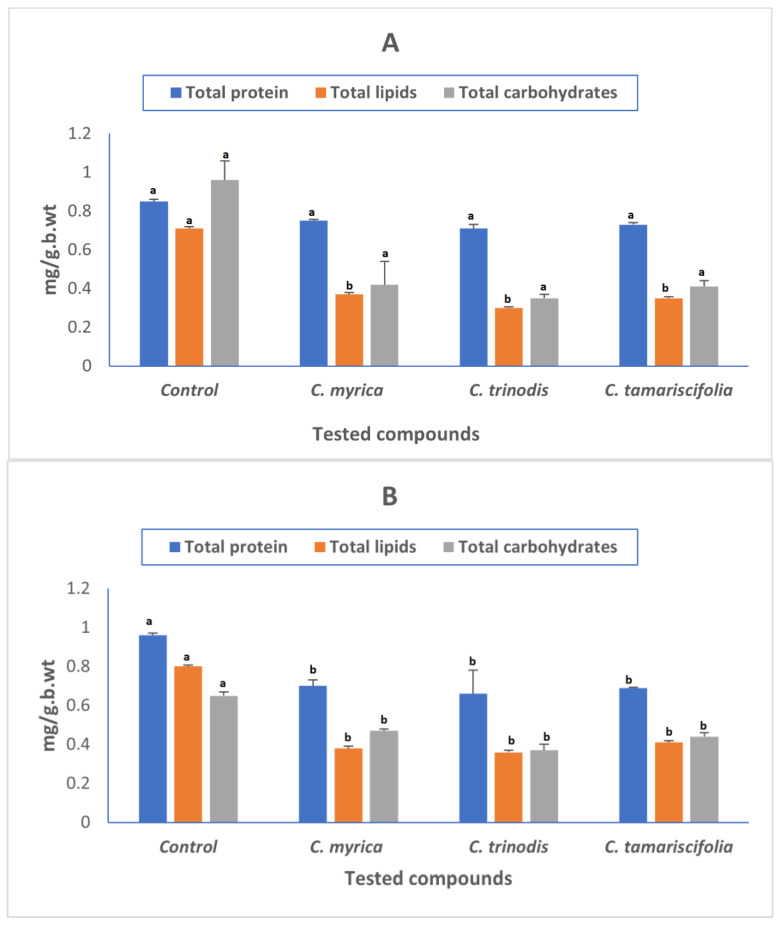
Effect of *Cystoseira myrica*, *C. trinodis*, and *C. tamariscifolia* extracts on biomolecules availability in *Culex pipiens* 3rd instar larvae 24 h (**A**), 48 h (**B**) post-treatment. (a,b) Different letters indicate that the mean is significantly different from control at *p* < 0.05.

**Figure 3 marinedrugs-21-00117-f003:**
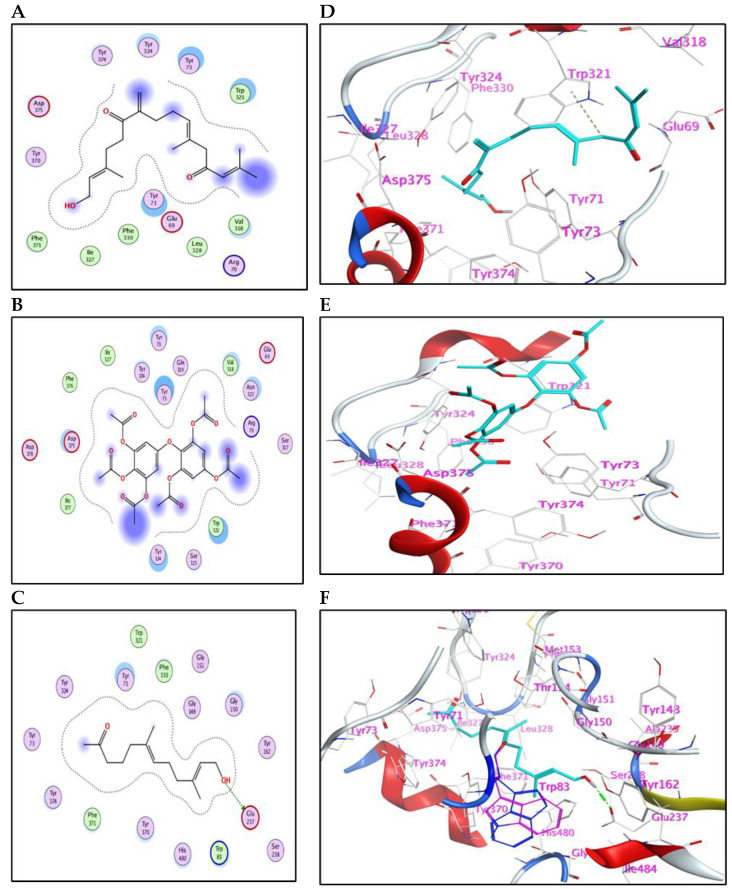
(**A**–**C**) 2D-binding interaction profile for compounds 3, 5, and 9, respectively, with the active pocket of the AChE enzyme. (**D**–**F**) In-depth 3D ligand-AChE interaction mode for compounds 3, 5, and 9, respectively.

**Figure 4 marinedrugs-21-00117-f004:**
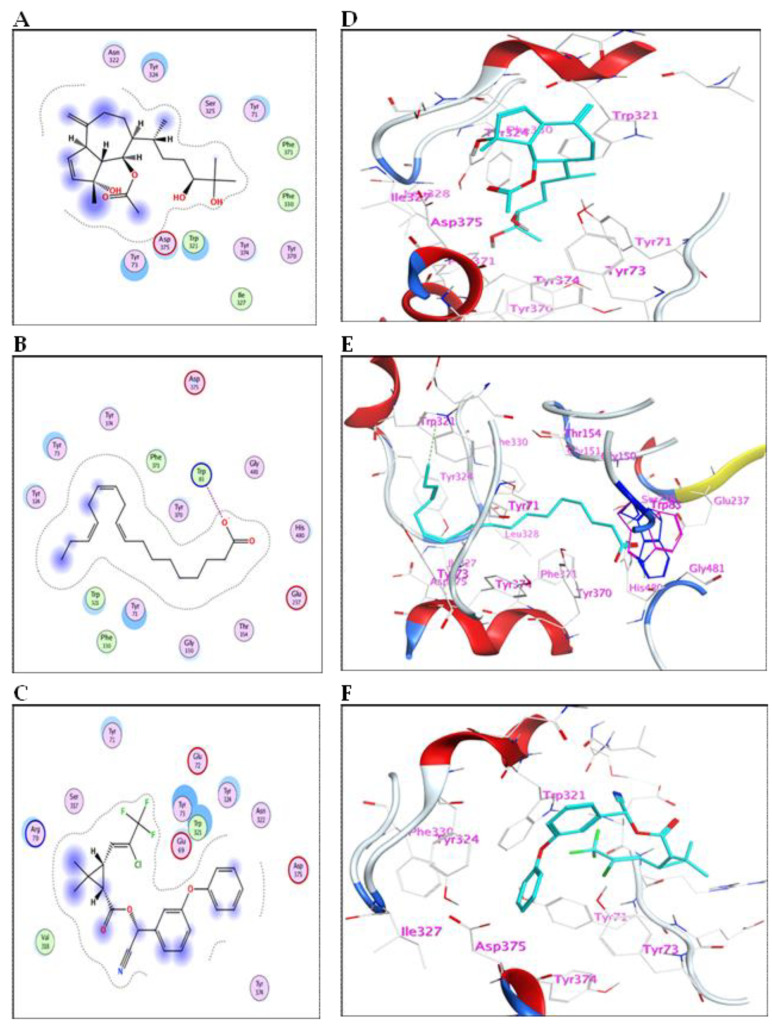
(**A**–**C**) 2D-binding interaction profile for compounds 10, 12, and λ-cyhalothrin, respectively, with the active pocket of AChE enzyme. (**D**–**F**) In-depth 3D ligand-AChE interaction mode for compounds 10, 12, and λ-cyhalothrin, respectively.

**Figure 5 marinedrugs-21-00117-f005:**
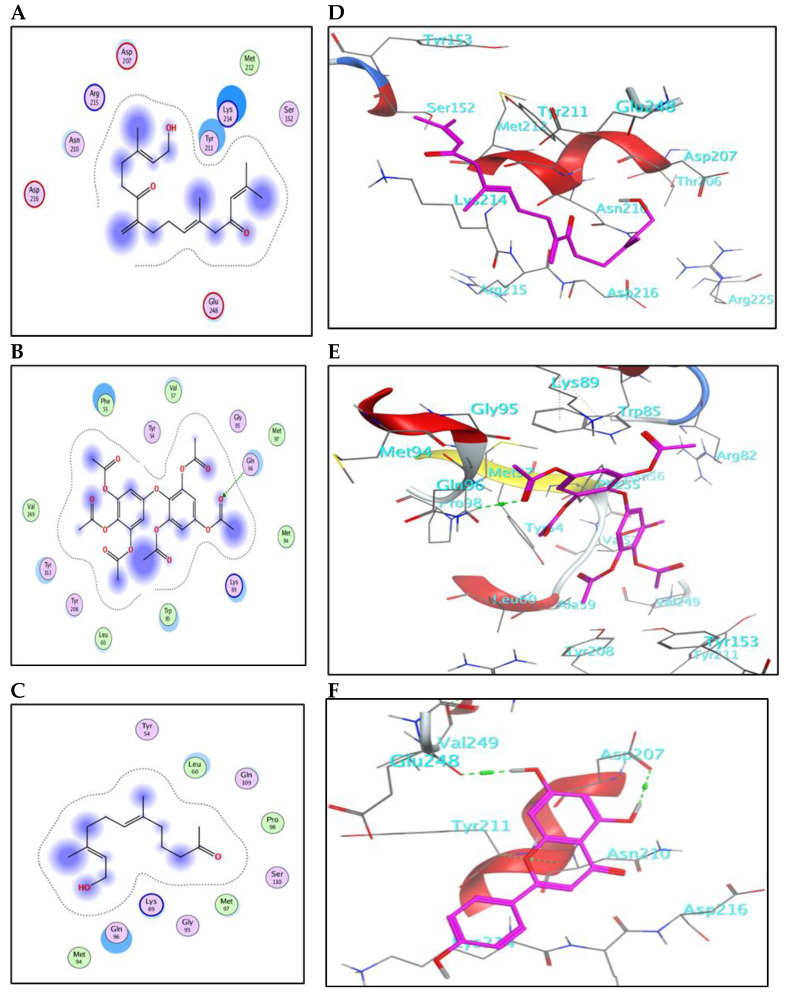
(**A**–**C**) The 2D-binding interaction profile for compounds 3, 5, and 9, respectively, with the pocket of GST protein. (**D**–**F**) In-depth 3D ligand-GST interaction mode for compounds 3, 5, and 9, respectively.

**Figure 6 marinedrugs-21-00117-f006:**
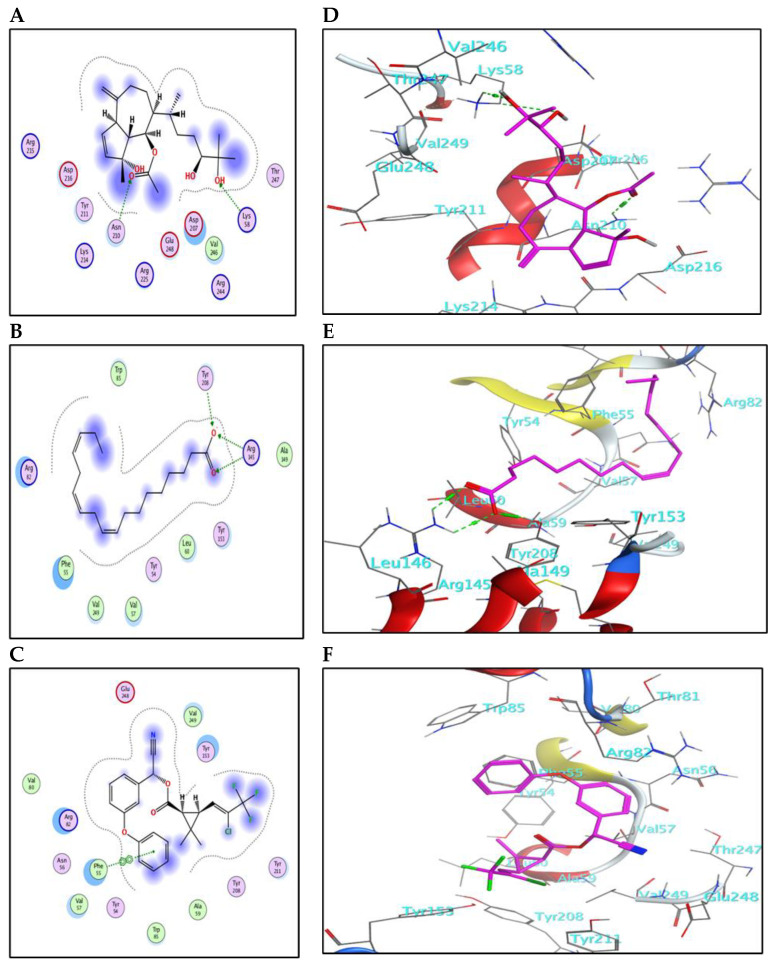
(**A**–**C**) The 2D-binding interaction profile for compounds 10, 12, and λ-cyhalothrin, respectively, with the pocket of GST protein. (**D**–**F**) In-depth 3D ligand-GST interaction mode for compounds 10, 12, and λ-cyhalothrin, respectively.

**Table 2 marinedrugs-21-00117-t002:** Susceptibility of *Culex pipiens* to *C. myrica*, *C. trinodis*, and *C. tamariscifolia* 24 h, 48 h, and 72 h post-treatment.

Conc. (ppm)	Percentage of Mortalities %								
	*C. myrica*			*C. trinodis*			*C. tamariscifolia*		
	24 h	48 h	72 h	24 h	48 h	72 h	24 h	48 h	72 h
0	0.0	0.0	0.0	0.0	0.0	0.0	0.0	0.0	0.0
50	18.66	26.66	34.66	10.66	18.33	22.66	5.3	13.33	21.33
100	44	52	60	24	36	41.33	21.33	32	44
150	60	68	72	52	60	69.33	60	68	76
200	86.66	80	92	74.66	82.66	88	76	82.66	88
250	90.66	94.66	98.66	81.33	90.66	92	90.66	93.33	97.33
LC_25_	64.51 (54.4–73.5)	51.99 (41.5–61.3)	44.74 (17.9–51.3)	95.02 (85–104)	75.17 (65.7–83.6)	60.27 (51–68.5)	86.96 (75.6–97)	68.65 (40.3–82)	60.33 (50.5–69)
LC_50_	105.06 (94.8–115)	91.09 (80–101.5)	75.63 (42.8–94.7)	135.08 (125.6–145)	113.42 (104–122.7)	94.66 (85.3–103.7)	138.71 (127.1–151)	111.51 (80.8–141)	98.15 (88.1–107.8)
LC_95_	345.06 (292–432.7)	357.67 (294.9–468.6)	272.09 (234.8–638.2)	318.54 (281–375.3)	309.33 (269.7–370.5)	284.62 (245.6–346.3)	433.21 (360.6–558)	364.01 (306.7–758.8)	321.55 (273.5–399.9)
* Slope ± SE	3.19 ± 0.28	2.77 ± 0.26	2.96 ± 0.27	4.41 ± 0.36	3.78 ± 0.30	3.44 ± 0.29	3.33 ± 0.29	3.20 ± 0.28	3.19 ± 0.28
* *X*^2^	7.38	5.89	10.62	5.46	7.43	5.57	6.16	8.43	7.69
*p*-value	0.06	0.11	0.0139	0.14	0.06	0.13	0.10	0.04	0.05

* Slope of the concentration-inhibition regression line ± standard error; * (*X*^2^) Chi-square value; LC values in ppm (95% C.I.) with the lower and the upper limit; LC values = Lethal concentrations values. 95% C.I. = Ninety-five percent confidence limit.

**Table 3 marinedrugs-21-00117-t003:** Quantitative analysis of glutathione-*S*-transferase and acetylcholinesterase activity at different time intervals.

Sample Tested	GST (UX10^3^/g.b.wt)		AChE (ug AchBr/min/g.b.wt)	
	24 h	48 h	24 h	48 h
Untreated	0.86 ± 0.03 ^a^	0.78 ± 0.06 ^a^	7.23 ± 0.26 ^a^	7.60 ± 0.37 ^a^
*C. myrica*	1.69 ± 0.005 ^b^	1.52 ± 0.08 ^b^	5.69 ± 0.4 ^b^	6.23 ± 0.33 ^b^
*C. trinodis*	1.70 ± 0.01 ^b^	1.54 ± 0.06 ^b^	6.16 ± 0.8 ^b^	6.26 ± 0.34 ^b^
*C. tamariscifolia*	1.72 ± 0.01 ^b^	1.56 ± 0.05 ^b^	6.26 ± 0.6 ^a^	6.36 ± 0.27 ^b^

Means bearing different scripts are significantly different from control at *p* < 0.05, Mean ± Standard error.

**Table 4 marinedrugs-21-00117-t004:** The binding data for major compounds identified in *C. myrica*, *C. trinodis*, and *C. tamariscifolia* methanol fraction with AChE and GST active sites.

Code	Compound Name	AChE Interaction		GST Interaction	
		BE ^a^ (∆G)	RMSD ^b^ Refine	BE (∆G)	RMSD ^b^ Refine
1	Compound 7	−5.02	1.34	−5.22	0.96
2	Myricetin	−6.17	1.41	−5.67	1.78
3	(2*E*, l0*E*)-1-hydroxy-6,13-diketo-7-methylene-3,11,15-trimethyl- hexadeca-2,l0,l4-triene	−7.18	1.97	−5.75	1.22
4	Luteolin-7-*O*-glucoside	−6.98	1.61	−6.36	1.18
5	Bifuhalol hexacetate	−7.26	1.84	−6.51	1.66
7	Cyanidin-3-*O*-glucoside	−6.80	1.17	−6.21	1.32
8	7,8-Methylenedioxycoumarin	−4.98	1.19	−4.42	1.72
9	Oxocrinol	−7.28	1.77	−5.23	1.42
10	Cystoseirol monoacetate	−6.86	1.84	−6.06	1.65
11	Apigenin	−5.59	0.51	−5.18	1.29
12	α-Linolenic acid	−7.97	1.89	−5.71	1.79
-	λ-Cyhalothrin	−6.57	1.16	−5.86	1.31

^a^ BE = Binding energy (Kcal/mol), ^b^ RMSD = Root-mean-square deviation.

## Data Availability

Data are available upon request.

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
