# Peer review of "Phytochemical Investigation of Three Cystoseira Species and Their Larvicidal Activity Supported with In Silico Studies"

_marinedrugs, 2023, doi:10.3390/md21020117_

Round 1

Reviewer 1 Report

See attached for comments

Reviewer 2 Report

1.      Abstract need restructuring; first sentence must be about the background and the problem; followed by aim of the study, methods used, results and a conclusion. revise accordingly

2.      In the introduction disease burden and disease prevalence, current drug discovery to treat malaria, especially from alternative sources must be focused and thus establish a need for the current work

3.      Space issues; methods 45 °C should be(45°C). Check throughout the text

4.      What was the rationale for testing the samples against the insects? is there any traditional use or the study design was random

5.      Phytochemical analysis; For confirmation of the identification, a co-injection of the chemical
standards of the tentatively identified compounds must be incorporated.

6.      Kindly confirm that any approval from ethical committee was required?

7.      How were the extracts applied? I mean how were their solutions prepared. Solution in organic solvents have inhibitory potentials against larvae

8.      What is the significance of checking the proteins, carbohydrates etc in the homogenates

9.      What is the relevance of cholinesterase enzyme inhibition here

10.  The docking study could have been performed against some vital enzyme or targets of the parasite, what is the relevance and significance here. Either change the title as it is only focused towards malaria and the other essays seems irrelevant here

Reviewer 3 Report

I suggest the publication of this manuscript after address the following issues:

i)                    Please add all MS spectra obtained and discussed. These spectra should be added as supplementary material.

ii)                   Please explain: Have you used the extracts in solution or powder? (Please add details in section 3.5).  If in solution, what volume was applied?

iii)                 Are extracts soluble in water?

iv)                 Please add a photo regarding the larvicidal bioassay containing the larvae and extracts evaluated (supplementary material).
